# Phenotypic and Genotypic Characterization of 171 Patients with Syndromic Inherited Retinal Diseases Highlights the Importance of Genetic Testing for Accurate Clinical Diagnosis

**DOI:** 10.3390/genes16070745

**Published:** 2025-06-26

**Authors:** Sofia Kulyamzin, Rina Leibu, Hadas Newman, Miriam Ehrenberg, Nitza Goldenberg-Cohen, Shiri Zayit-Soudry, Eedy Mezer, Ygal Rotenstreich, Iris Deitch, Daan M. Panneman, Dinah Zur, Elena Chervinsky, Stavit A. Shalev, Frans P. M. Cremers, Dror Sharon, Susanne Roosing, Tamar Ben-Yosef

**Affiliations:** 1The Ruth & Bruce Rappaport Faculty of Medicine, Technion-Israel Institute of Technology, Haifa 3109601, Israel; sofia.itskov@campus.technion.ac.il (S.K.); nitza.cohen@b-zion.org.il (N.G.-C.); emezer@technion.ac.il (E.M.); stavit_sh@clalit.org.il (S.A.S.); 2Department of Ophthalmology, Rambam Health Care Campus, Haifa 3109601, Israel; rinaleibu@gmail.com; 3Division of Ophthalmology, Tel Aviv Sourasky Medical Center, Tel Aviv 6423906, Israel; hadasng@gmail.com (H.N.); dinahzur@gmail.com (D.Z.); 4Department of Ophthalmology, Schneider Children’s Medical Center of Israel, Petach Tikva 4920235, Israel; miriberg@gmail.com; 5Gray Faculty of Medical and Health Sciences, Tel Aviv University, Tel Aviv 6997801, Israel; shirizayit@gmail.com (S.Z.-S.); ygal.rotenstreich@sheba.health.gov.il (Y.R.); 6Department of Ophthalmology, Bnai Zion Medical Center, Haifa 3339419, Israel; 7Department of Ophthalmology, Rabin Medical Center, Petach Tikva 4941492, Israel; iris.deitch@gmail.com; 8Goldschleger Eye Institute, Sheba Medical Center, Tel Hashomer 5262100, Israel; 9Department of Human Genetics, Radboud University Medical Center, 6525 Nijmegen, The Netherlands; d.m.panneman@umcutrecht.nl (D.M.P.); frans.cremers@radboudumc.nl (F.P.M.C.); susanne.roosing@radboudumc.nl (S.R.); 10Genetics Institute, Emek Medical Center, Afula 1834111, Israel; elana_ch@clalit.org.il; 11Division of Ophthalmology, Hadassah Medical Center, Faculty of Medicine, The Hebrew University of Jerusalem, Jerusalem 9112001, Israel; dror.sharon1@mail.huji.ac.il

**Keywords:** inherited retinal disease, syndrome, retina, KATNIP, PEX6, TUBB4B, Usher syndrome

## Abstract

Background: Syndromic inherited retinal diseases (IRDs) are a clinically and genetically heterogeneous group of disorders, involving the retina and additional organs. Over 80 forms of syndromic IRD have been described. Methods: We aimed to phenotypically and genotypically characterize a cohort of 171 individuals from 140 Israeli families with syndromic IRD. Ophthalmic examination included best corrected visual acuity, fundus examination, visual field testing, retinal imaging and electrophysiological evaluation. Most participants were also evaluated by specialists in fields relevant to their extra-retinal symptoms. Genetic analyses included haplotype analysis, homozygosity mapping, Sanger sequencing and next-generation sequencing. Results: In total, 51% of the families in the cohort were consanguineous. The largest ethnic group was Muslim Arabs. The most common phenotype was Usher syndrome (USH). The most common causative gene was *USH2A*. In 29% of the families, genetic analysis led to a revised or modified clinical diagnosis. This included confirmation of an atypical USH diagnosis for individuals with late-onset retinitis pigmentosa (RP) and/or hearing loss (HL); diagnosis of Heimler syndrome in individuals with biallelic pathogenic variants in *PEX6* and an original diagnosis of USH or nonsyndromic RP; and diagnosis of a mild form of Leber congenital amaurosis with early-onset deafness (LCAEOD) in an individual with a heterozygous pathogenic variant in *TUBB4B* and an original diagnosis of USH. Novel genotype–phenotype correlations included biallelic pathogenic variants in *KATNIP*, previously associated with Joubert syndrome (JBTS), in an individual who presented with kidney disease and IRD, but no other features of JBTS. Conclusions: Syndromic IRDs are a highly heterogeneous group of disorders. The rarity of some of these syndromes on one hand, and the co-occurrence of several syndromic and nonsyndromic conditions in some individuals, on the other hand, complicates the diagnostic process. Genetic analysis is the ultimate way to obtain an accurate clinical diagnosis in these individuals.

## 1. Introduction

Inherited retinal diseases (IRDs) are a clinically and genetically heterogeneous group of diseases, causing visual loss due to the abnormal development, dysfunction or degeneration of photoreceptors or the retinal pigment epithelium [1,2]. The most common form of IRD is retinitis pigmentosa (RP), also known as rod-cone dystrophy. Other forms include cone and cone–rod dystrophy (CD/CRD), inherited macular dystrophies and more. IRDs are one of the most genetically heterogeneous group of disorders in humans. They can be inherited as autosomal recessive (AR), autosomal dominant (AD) or X-linked. Mitochondrial and digenic patterns of inheritance have also been described. To date, over 320 genes have been implicated in IRD (RetNet at https://sph.uth.edu/Retnet/, accessed on 1 May 2025).

Most IRD cases are nonsyndromic (isolated; involving only ophthalmic manifestations); nevertheless, over 80 forms of syndromic IRD have been described [3]. These syndromes are associated with variants in approximately 200 genes. Many forms of syndromic IRD exhibit marked phenotypic variability, and some genes are involved in both syndromic and nonsyndromic IRD forms, depending on the nature and combination of the causative allele/s. The most common form of syndromic IRD is Usher syndrome (USH) (prevalence of 4 to 17 cases per 100,000 individuals), an AR disorder characterized by the combination of RP and sensorineural hearing loss (SNHL), with or without vestibular areflexia. USH is traditionally classified into three major subtypes (USH1-3), the most severe of which is USH1, characterized by congenital profound SNHL, precluding normal development of speech, vestibular areflexia and onset of RP at the first decade of life. The most common form of USH worldwide is USH2, characterized by early-onset, moderate-to-severe, stable SNHL, most pronounced at the higher frequencies, and postpubertal onset of RP, with normal vestibular function. USH3 includes a variable phenotype, characterized by post-lingual progressive SNHL (usually diagnosed in the first decade), varying levels of vestibular dysfunction and average onset of RP in the second decade of life, although later onset may occur [4]. Nevertheless, in recent years, several genes have been reported to be associated with a clinical subtype called “atypical USH” that does not meet the canonical criteria for the three recognized USH subtypes [4].

Like USH, many forms of syndromic IRD are defined as ciliopathies: a group of genetic diseases caused by mutations in genes associated with the structure and function of primary cilia [5]. In the retina, the primary cilium serves as the connecting bridge between the photoreceptor inner and outer segments; its dysfunction leads to impaired photoreceptor function and progressive degeneration. Additional organs which are commonly affected in ciliopathies, besides the retina, are the central nervous system, inner ear, skeleton, kidney and liver [3]. One of the major forms of syndromic IRD that belongs to the ciliopathy group is Bardet–Biedl syndrome (BBS) (estimated prevalence of 1 in 100,000 to 1 in 160,000 individuals), an AR multisystemic disease characterized by retinal dystrophy, postaxial polydactyly, renal disease, intellectual disability and truncal obesity [6,7]. Another ciliopathy is Joubert syndrome (JBTS) (estimated prevalence of 1 in 80,000 to 1 in 100,000 individuals), an AR severe disorder classically characterized by three primary findings: a distinctive cerebellar and brainstem malformation called the molar tooth sign, hypotonia and cognitive impairment. These may be accompanied by breathing abnormalities, ataxia, renal disease, genitourinary abnormalities, retinal dystrophy, ocular colobomas, occipital encephalocele, hepatic fibrosis, polydactyly and endocrine abnormalities [8,9]. Senior–Løken syndrome (estimated prevalence of 1 in 1,000,000 individuals) is an AR condition characterized by the combination of retinal dystrophy and nephronophthisis [10].

A rare AD ciliopathy is Leber congenital amaurosis with early-onset deafness (LCAEOD), caused by pathogenic variants in the *TUBB4B* gene [11]. *TUBB4B* encodes for beta-tubulin 4b. Microtubules are dynamic polymeric structures consisting of heterodimers of alpha-tubulins and beta-tubulins, such as TUBB4B, which function in mitosis, intracellular transport, neuron morphology, and ciliary and flagellar motility. Of note, some pathogenic variants of *TUBB4B* are associated with a distinct phenotype, primary ciliary dyskinesia (PCD), a disorder mainly affecting the respiratory system [12]. Some patients with heterozygous pathogenic variants of *TUBB4B* are affected by both conditions simultaneously [12].

Another major group of IRDs is inborn errors of metabolism (IEM). IEMs are genetic disorders leading to the failure of carbohydrate metabolism, protein metabolism, fatty acid oxidation or glycogen storage. Many IEMs present with neurologic symptoms [13]. The retina is considered an extension of the brain. Therefore, neurodegeneration resulting from IEMs often involves retinal degeneration as well. One form of syndromic IRD that belongs to the IEM group is peroxisomal biogenesis disorders (PBDs) (aggregated frequency of approximately 1 in 50,000 live births) [14]. Two disorders included in the PBD group are peroxisomal biogenesis disorder 4A (PBD4A, Zellweger syndrome), a multiple congenital anomaly syndrome, and Heimler syndrome 2 (HMLR2), a less severe phenotype involving IRD, SNHL, dental anomalies (amelogenesis imperfecta) and nail abnormalities. Both PBD4A and HMLR2 are caused by biallelic pathogenic variants in the *PEX6* gene [15,16].

Genotypic and phenotypic analyses of individuals with syndromic IRD from various ethnic backgrounds have been previously reported as part of large cohorts of individuals with both syndromic and non-syndromic IRD [17,18,19,20,21,22]. Other studies focused on individuals with one specific form of syndromic IRD (such as JBTS or BBS) [23,24]. One study reported a cohort of 100 Spanish individuals with syndromic IRD only but excluded USH patients [25]. Here, we describe phenotypic and genotypic characterization of a cohort of 171 Israeli patients with various forms of syndromic IRD.

## 2. Materials and Methods

### 2.1. Subjects

A total of 1435 Israeli families were recruited between the years 2004 and 2025 to our ongoing study on the genetic basis for IRD, conducted at the Rappaport Faculty of Medicine in the Technion-Israel Institute of Technology (the term “family” refers to a nuclear family; thus, each proband and his first-degree relatives, if recruited, were defined as a family). In 891 of these families, the underlying genetic cause was identified. Of these genetically solved families, 140 (16%) segregate a syndromic IRD type. These families, including 171 affected individuals, were defined as the syndromic IRD cohort described in this study (Figure 1A and Appendix A). Of the 544 unsolved families, 49 were defined as syndromic or possibly syndromic. Thirty of these families were analyzed by whole exome sequencing (WES) or whole genome sequencing (WGS) (Appendix A). In three of them, one heterozygous pathogenic allele was identified in a syndromic IRD-causative gene compatible with their phenotype and inherited in an AR mode. A second heterozygous pathogenic allele in the same gene was not identified, despite the fact that two of these families were analyzed by WGS. Some of the families and/or pathogenic variants included in this study were previously reported by us, but not in the context of a syndromic IRD cohort [26,27,28,29,30,31,32,33,34,35,36,37] (Appendix A).

The study was performed according to the tenets of the Declaration of Helsinki and was approved by the Institutional Review Boards (IRBs) of all participating institutions (see details in the Institutional Review Board Statement). Informed consent was obtained from all participants or their parents, as well as from unaffected family members.

### 2.2. Clinical Evaluation

Ophthalmic diagnoses were made based on complete ophthalmic examination, including best corrected visual acuity (BCVA), fundus examination, visual field testing, retinal imaging by optical coherence tomography (OCT) and fundus autofluorescence (FAF) and electrophysiological evaluation by full-field electroretinography (ffERG) and/or multi-focal electroretinography (mfERG). Most participants were also evaluated by specialists in fields relevant to their symptoms, including neurology, otolaryngology, nephrology, hepatology and/or medical genetics.

### 2.3. Genetic Analyses

Testing strategies used in each family are listed in Appendix A. Index patients from 59 families were tested for founder mutations prevalent in their ethnic group or village, by PCR amplification with specifically designed primers, followed by direct sequencing. Index patients from 56 families were subjected to WES as previously described [38]. Index patients from two families were subjected to WGS as previously described [28]. Index patients from six families were tested by targeted next generation sequencing (TNGS) of 108 or 113 known IRD genes using the Molecular Inversion Probes technique [39,40]. Eight families were analyzed by haplotype analysis, followed by PCR-amplification and Sanger sequencing of relevant candidate genes, as previously described [41]. Affected individuals from six families were analyzed by homozygosity mapping, followed by PCR-amplification and Sanger sequencing of relevant candidate genes, as previously described [35]. In three index patients, specific candidate genes were targeted based on the phenotype, and all their coding exons were PCR-amplified with or without consecutive Sanger sequencing.

### 2.4. Bioinformatics

Public databases used to determine population allele frequencies included gnomAD [42], ESP6500 [43], TOPMed BRAVO [44] and GME Variome [45].

The pathogenicity of missense variants was evaluated based on the Franklin by Genoox (https://franklin.genoox.com/) aggregated prediction score, which is based on scores obtained by multiple prediction tools including REVEL [46], MutationAssessor/r3 [47], SIFT [48], Polyphen-2 [49], MutationTaster 2021 [50], FATHMM (v2.3) [51], DANN (v3.19) [52], MetaLR [53], PrimateAI (v1.0) [54] and BayesDel (v1) [55]. The putative effect of certain variants on splicing was evaluated based on the following prediction tools: SpliceAI (v1.3.1) [56], dbscSNV_AdaBoost[57] and dbscSNV_RandomForest [58] (v1.1). Analysis of the evolutionary conservation of KATNIP Ser1571 was performed with The ConSurf Server (https://consurf.tau.ac.il/consurf_index.php) [59].

## 3. Results

### 3.1. Characteristics of the Syndromic IRD Cohort

Seventy-two of the families in the syndromic IRD cohort (51%) are consanguineous. Muslim Arabs constitute the largest ethnic group in the cohort (36%), followed by Ashkenazi Jews (20%), North African Jews (11%), Oriental Jews (11%), Jews of mixed ancestry (7%), Yemenite Jews (6%), Druze (5%), Christian Arabs (2%) and others (Figure 1B).

Individuals in this cohort were affected by 24 distinct syndromic IRD phenotypes, caused by 96 pathogenic variants in 41 different genes. Of these variants, 81 have been previously reported, while 15 are novel. The most common phenotype in the cohort was USH (65% of families), followed by BBS (6%) and hypotrichosis with juvenile macular dystrophy (4%) [60] (Figure 1C). Among families with USH, the most common type was USH2 (44%). The most common causative gene in the entire cohort was *USH2A* (NM_206933.2) (31%), followed by *MYO7A* (NM_000260.3) (10%) and *CLRN1* (NM_001195794.1) (6%) (Figure 1D). The most common causative variant in the entire cohort was *USH2A*: c.236_239dup (Figure 1E), present in 20 alleles of 12 probands.

**Figure 1 genes-16-00745-f001:**
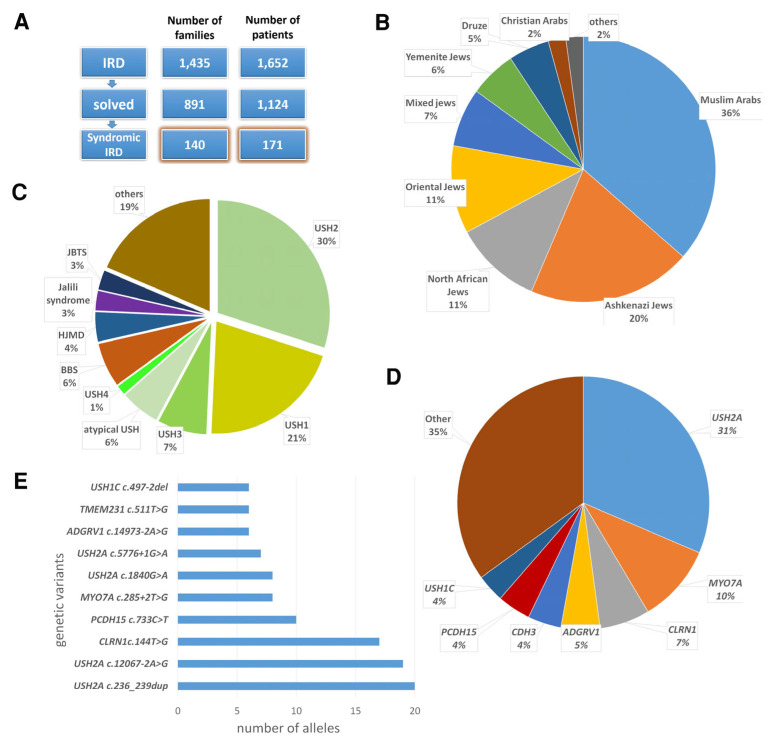
Phenotypic and genotypic characteristics of the syndromic IRD patient cohort. (**A**) Assembly of the syndromic IRD cohort. (**B**) Ethnic distribution of the cohort. (**C**) Phenotypic distribution of the cohort. BBS, Bardet–Biedl syndrome; HJMD, hypotrichosis with juvenile macular dystrophy; JBTS, Joubert syndrome; USH, Usher syndrome. (**D**) Distribution of mutated genes in the cohort. (**E**) The most common causative variants identified in the cohort.

### 3.2. Revised Clinical Diagnosis Following Genetic Analysis

In 41 of the families (29%), genetic analysis led to a revised or modified clinical diagnosis (Appendix A). Most of these families (70%) were analyzed by WES. For example, in several individuals with a combination of RP and HL, late onset of one or both of these features led to uncertainty regarding a possible diagnosis of USH, since the phenotype did not fit any of the major USH types (USH1-3). Consequently, it was unclear if RP and HL in these cases are part of a syndrome or arose independently. Genetic analysis led to the identification of pathogenic variants in USH-related genes (mainly *USH2A*) in each of these cases, thus confirming a diagnosis of atypical USH (Appendix A and Table 1).

Interestingly, biallelic pathogenic variants in the *USH2A* gene were identified in two unrelated individuals from families TB120 and TB891, with an USH1 diagnosis due to congenital profound SNHL. One individual was homozygous for the common c.236_239dup allele, while the other was compound heterozygous for c.236_239dup and c.1840G>A; p.(Gly614Arg). Of note, the c.236_239dup allele was identified in homozygosity in seven additional individuals, six of them diagnosed with classic USH2 and one with atypical USH (Appendix A).

In individuals from two families, originally diagnosed with USH or with nonsyndromic RP, the identification of pathogenic variants in *PEX6* (NM_000287) led to a revised clinical diagnosis. The participant from family TB725 was referred to clinical and genetic evaluation due to IRD with onset at the age of 30 y. WES revealed a likely homozygous variant in *PEX6*, c.1802G>A; p.(Arg601Gln). This variant is rare (gnomAD aggregated allele frequency (AF) of 0.3%) and predicted pathogenic (Franklin aggregated prediction score = 0.86). It has been previously reported in individuals with either HMLR2 or PBD4A [15,61,62]. The genetic finding led to focused clinical re-evaluation of this participant, which revealed late onset mild HL and a very mild dental abnormality. These clinical findings led to a revised clinical diagnosis of HMLR2. It should be noted that DNA from the parents of this individual was not available for segregation analysis, and that we did not perform a focused analysis to rule out the possibility that this individual is actually compound heterozygous for p.(Arg601Gln) and for a complete or partial deletion of the *PEX6* gene. Such a deletion would be considered a null allele, and given the very mild phenotype of this patient, this option is less likely.

In the participant from family TB1003, the combination of congenital SNHL and late-onset RP led to a possible diagnosis of USH. WES revealed two heterozygous variants in *PEX6*, c.2094+2T>C and c.2534T>C;p.(Ile845Thr). Both variants are rare (c.2094+2T>C not present in public databases; c.2534T>C gnomAD aggregated AF of 0.003%), and they were both reported previously in compound heterozygosity in an Israeli individual with an USH-like phenotype [63]. A repeated clinical evaluation of our participant revealed that she had severe dental problems in childhood, probably amelogenesis imperfecta. Her clinical diagnosis was, thus, amended to HMLR2.

Diagnosis was also revised in an individual from family TB1134. This participant had SNHL since childhood and received a cochlear implant at the age of 36 y. She was diagnosed with RP at the age of 51 y. She also experienced multiple respiratory infections and had a family history of severe lung disease in her father. The combination of childhood-onset SNHL and RP led to a diagnosis of USH. WES revealed a heterozygous variant in *TUBB4B* (NM_006088.6), c.16C>T; p.(His6Tyr). This variant is rare (not present in public databases) and predicted pathogenic (Franklin aggregated prediction score = 0.7). To date, 33 individuals harboring heterozygous pathogenic variants of *TUBB4B* have been reported (including the current report). Of them, 30 (91%) had HL, 24 (73%) had IRD, 14 (42%) had PCD/recurrent respiratory infections, six (18%) had hydrocephalus, three had kidney disease, three had congenital heart disease and three had skeletal abnormalities (9% each) [11,12,64,65,66,67,68,69] (Figure 2).

### 3.3. Co-Occurrence of Syndromic IRD and Additional Phenotypes

The co-occurrence of non-syndromic IRD and additional non-syndromic phenotypes, leading to a false diagnosis of syndromic IRD, is not uncommon [70]. However, in some of the participants, we identified the co-occurrence of syndromic IRD with additional syndromic or non-syndromic conditions. For example, in family TB939, the proband had a combination of RP and HL (leading to an initial diagnosis of USH), as well as truncal obesity, hypoplastic kidney and diabetes mellitus. WES revealed homozygous pathogenic variants in two genes: *BBS1* (NM_024649.4) c.437G>C; p.(Arg146Pro) (underlying BBS [71]) and *CABP2* (NM_016366) c.419T>G; p.(Met140Arg) (underlying nonsyndromic HL [72]). The diagnosis was, thus, changed to BBS and HL. In family TB1199, the proband had multiple symptoms, including RP, HL, intellectual disability, short stature, truncal obesity and phalangeal abnormalities. WES revealed a homozygous pathogenic variant in *ADGRV1* (NM_032119.3) (c.9335_9336inv; p.(Phe3112*)), accounting for RP and HL and establishing a diagnosis of USH2 [73]. In addition, the proband was heterozygous for a pathogenic variant of the *SMARCA2* gene (NM_003070.5) (c.2657A>G; p.(Asn886Ser)). This variant is rare (not present in gnomAD) and predicted pathogenic (Franklin aggregated prediction score = 0.98). Heterozygous pathogenic variants in *SMARCA2* are associated with Blepharophimosis-impaired intellectual development syndrome or with Nicolaides–Baraitser syndrome [74,75]. Both syndromes are inherited as AD, and they have partially overlapping phenotypes, which include intellectual disability, short stature and phalangeal abnormalities. This patient is, therefore, simultaneously affected by two distinct syndromes.

### 3.4. New Genotype–Phenotype Correlation: Biallelic Pathogenic Variants in KATNIP in an Individual with a JBTS26-Related Ciliopathy

Family TB398 is a non-consanguineous family of mixed Jewish origin. The proband is a female who was seen by us at the age of 24 y. She had normal growth and development, and normal cognitive function. At the age of 7 y, she was diagnosed with type 2 glomerulonephritis, and subsequently underwent two kidney transplantations, at the ages of 9 and 16 y. At the age of 15 y, she developed progressive SNHL, which was attributed to antibiotic toxicity. At the age of 22 y she was diagnosed with high intracranial pressure, which was treated pharmacologically (Diamox) and surgically (lumbar punctures). At the same age, she started experiencing visual disturbances with severe photophobia. BCVA was 6/7 in both eyes. Funduscopy revealed salt and pepper pigmentation of the retinal pigmented epithelium in the fovea and macula, surrounded by white dots and patches extending to the periphery. OCT revealed irregularity of the external band. ffERG indicated reduced responses under both photopic and scotopic conditions. mfERG demonstrated a reduced foveal peak with preserved peripheral responses. These findings led to a differential diagnosis of CRD versus macular dystrophy with peripheral involvement. Retinal involvement secondary to the kidney disease was also considered. WES revealed two rare heterozygous variants in the *KATNIP (KIAA0556)* gene (NM_015202.4): c.49C>T; p.(Arg17*) and c.4711A>G; p.(Ser1571Gly). c.49C>T is a nonsense variant, which is not present in gnomAD but was reported in ClinVar as pathogenic. c.4711A>G is a missense variant with a gnomAD aggregated AF of 0.008%. It affects a highly conserved amino acid (Appendix A) and is predicted pathogenic (Franklin aggregated prediction score = 0.8), but it was reported in ClinVar as a VUS. Pathogenic variants in *KATNIP* have been associated with JBTS, a severe phenotype compatible with retinal degeneration and kidney disease, and usually involving developmental delay and intellectual disability [76]. HL has not been reported as part of the JBTS-associated phenotype. Of note, the proband was also found to be heterozygous for a likely-pathogenic variant in *POLG* (Appendix A). Pathogenic variants in *POLG* are mostly associated with a spectrum of AR mitochondrial conditions. However, AD inheritance has also been described, mainly associated with progressive external ophthalmoplegia, which may be accompanied by additional neurological features, including progressive SNHL [77]. The proband did not present with ophthalmoplegia, and whether her HL is caused by this *POLG* variant remains unclear.

## 4. Discussion

Over 80 forms of syndromic IRD have been described to date [3], over 20 of which were diagnosed in individuals from the cohort described here. The most common phenotype in the cohort was USH (and specifically USH2), and the most common causative gene was *USH2A*. These findings are similar to reported findings in other IRD cohorts worldwide [17,18,19,20,21,22]. Nevertheless, while this study confirms some previously reported findings, it also adds new insights, as detailed below.

Analysis of the USH cases in our cohort emphasizes the phenotypic variability associated with pathogenic variants in *USH2A*. While most individuals with biallelic pathogenic variants in *USH2A* are diagnosed with USH2 [78] or with nonsyndromic RP [79], some have atypical USH, with late onset of HL and/or RP (Table 1) (reviewed in [4]). In contrast, we also identified two individuals with biallelic pathogenic variants in *USH2A* who were phenotypically classified as USH1. Such cases have been reported previously but are quite rare [80]. Interestingly, one of these individuals was homozygous for a common allele in the Israeli population, c.236_239dup, which is usually associated with an USH2 phenotype. This finding suggests the involvement of additional genetic and/or environmental modifying factors.

Another interesting finding is the tendency to diagnose individuals with the combination of HL and RP with USH, while overlooking additional phenotypic features, which might point to the actual diagnosis. While USH is the most prevalent form of syndromic IRD and the most common cause for genetic deaf-blindness, at least 11 other syndromes involving RP and HL have been described [3]. One of them is Heimler syndrome. Our findings suggest that this syndrome is under-diagnosed, and some patients may obtain a diagnosis of USH or even nonsyndromic RP, if the dental anomalies are overlooked. Similarly, the participant with a heterozygous *TUBB4B* pathogenic variant was also diagnosed with USH, due to the combination of SNHL and RP. However, the very late onset of RP in this case, and the history of recurrent respiratory infections, suggested a different diagnosis, as, indeed, revealed by genetic testing.

To date, pathogenic variants in *TUBB4B* have been associated with two distinct AD phenotypes, LCAEOD and PCD, which may overlap in some patients [11,12]. The individual described here presented with early-onset SNHL, IRD and recurrent respiratory infections, which is in agreement with previous reports. However, our findings in this patient and the retrospective analysis of all patients reported to date demonstrate that pathogenic variants in *TUBB4B* are not necessarily associated with two distinct disorders, LCAEOD or PCD, but rather with a phenotypic spectrum, with different combinations of features found in both diseases. This conclusion was previously implied [12] but is further established in the current study. Moreover, we argue that the name LCAEOD is misleading. LCA is mostly characterized with severe visual impairment, which is either congenital or diagnosed within the first year of life [81]. Of the LCAEOD-individuals reported to date, only seven were reported to have LCA and/or diagnosed with visual impairment by the age of 1 y. Seven had early-onset IRD (ages 1–6 y), while five (including the individual described here) had adult-onset (ages 25–51 y) [11,12,64,65,66,67,68,69]. In addition, while most individuals presented with LCA or RP, at least two had CRD [64,69]. *TUBB4B*-associated disease is, therefore, a ciliopathy, which primarily affects the ear, the retina and the respiratory system, but it might affect additional systems as well; the associated retinal phenotype can appear from birth to late adulthood, and present as LCA, RP or CRD.

JBTS is an AR multisystemic ciliopathy, exhibiting both intra- and interfamilial clinical variation. At least 40 causative genes have been identified; one of them is *KATNIP* (underlying JBTS26) [76]. Biallelic pathogenic variants in *KATNIP* have been reported to date in only a few individuals worldwide, all diagnosed with JBTS [76,82,83,84,85,86]. Clinical data were provided for 10 of 11 reported individuals; nine of them were reported to have cognitive impairment, mostly severe. Ophthalmic symptoms in some of these individuals included oculomotor apraxia, nystagmus and ptosis; two siblings from one family had cone dystrophy/reduced cone function [86]. The individual reported here is heterozygous for two rare variants of *KATNIP*: a nulll variant (p.(Arg17*)) and a missense variant (p.(Ser1571Gly)), which is predicted pathogenic. She had normal cognitive function, with no hypotonia and no known brain abnormalities. Her main features were renal disease and RD. Renal disease was not previously reported in individuals with *KATNIP* pathogenic variants. Nevertheless, renal disease was found in individuals with JBTS due to pathogenic alleles in other genes [8,9]. Moreover, according to the Human Protein Atlas (www.proteinatlas.org), *KATNIP* is expressed in the kidney. The association of intracranial hypertension in a ciliopathy is a rare occurrence [87]. While cilia are involved in the central nervous system, and ciliary dysfunction may contribute to intracranial hypertension [88,89], the etiology of intracranial hypertension in this case may be also due to renal transplantation and/or post-renal transplant medications [90]. Nevertheless, this patient presents a milder JBTS26-associated phenotype, which extends the phenotypic spectrum associated with biallelic pathogenic variants of *KATNIP*.

Of the 140 families in the syndromic IRD cohort, 59 (42%) were genetically diagnosed by testing for common founder mutations and 56 (40%) by WES (Appendix A). This indicates that in populations with strong founder effects, such as the Israeli population [91], the Dutch population [92], the Finnish population [93] and others, founder testing is a cost-effective approach for genetic diagnosis of inherited conditions, including syndromic IRD. WES is a highly effective approach for genetic analysis of syndromic IRD in all populations. While TNGS (gene panels) is commonly used in clinical settings, in Israel and in other Western countries, most commercially available IRD gene panels include mainly causative genes for nonsyndromic IRD (some of which, like *USH2A*, may also be associated with syndromic IRD). Given the high degree of genetic and clinical heterogeneity associated with syndromic IRD, WES is a broader approach with a higher chance to identify the causative variants, including in genes that are rare and hard to predict based on the phenotype alone. Patients who remain unsolved following WES should be subjected to more advanced genomic technologies, including WGS, long-read sequencing and optical genome mapping, which may identify variants in non-coding regions and complex structural aberrations [94,95,96].

In summary, the current study describes a relatively large cohort of 171 Israeli patients with syndromic IRD. While previous studies mainly reported findings on syndromic IRD as part of large IRD cohorts, or focused on specific types of syndromic IRD, our study included patients with various forms of syndromic IRD and focused on the unique challenges associated with clinical diagnosis of such patients. The availability of genetic data and comprehensive clinical data for these patients allowed us to perform a thorough genotype–phenotype correlation. This led to interesting outcomes, including a better definition of the *TUBB4B*-associated phenotype and broadening of the phenotypic spectrum associated with *KATNIP* pathogenic variants. Nevertheless, the application of WES and WGS on additional individuals with syndromic or possibly syndromic IRD from our general IRD cohort could increase the number of participants in the study. Also, it should be noted that most of the study participants were from Northern and Central Israel, while only a few were from Southern Israel and the Jerusalem area. Consequently, the relative frequencies of various phenotypes, genes and alleles in this cohort do not necessarily represent the entire Israeli population.

## 5. Conclusions

Syndromic IRD is a highly heterogeneous group of disorders, both clinically and genetically. The rarity of some of these syndromes, on one hand, and the co-occurrence of several syndromic and nonsyndromic conditions in some individuals, on the other hand, complicate the diagnostic process. This study further emphasizes these challenges and demonstrates how genetic analysis, combined with detailed phenotypic assessment, is the ultimate way to obtain an accurate clinical diagnosis in these individuals.

## Figures and Tables

**Figure 2 genes-16-00745-f002:**
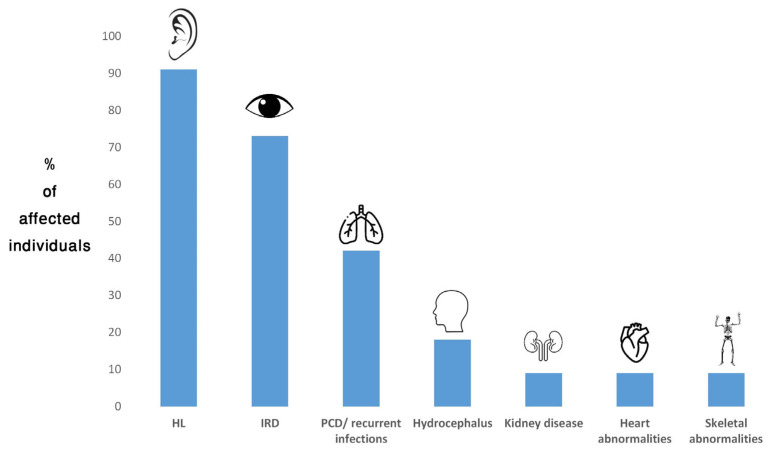
Frequency of *TUBB4B*-associated phenotypes in affected individuals reported to date. HL, hearing loss; IRD, inherited retinal disease; PCD, primary ciliary dyskinesia.

**Table 1 genes-16-00745-t001:** Retinal, auditory and genetic characteristics of individuals in whom genetic analysis confirmed a diagnosis of atypical USH.

Family- Individual (Sex)	Origin	Age of Onset of RP (Y)	Age of Onset of HL (Y)	HL Characteristics (Age-Y)	Gene	Pathogenic Variant/s
TB441-1 (M)	ASH/non-Jewish	18	30	Mild (30)	*USH2A*	c.4174G>T; p.(Gly1392*) het
						c.12575G>A; p.(Arg4192His) het
TB592-1 (M)	NAJ	50	55	Mild–moderate (59)	*USH2A*	c.1267-2A>G hom
TB639-1 (M)	OJ	42	30	Moderate (48)	*USH2A*	c.236_239dup; p.(Gln81Tyrfs*28) hom
TB699-1 (M)	NAJ	40	40	Moderate (59)	*USH2A*	c.1000C>T; p.(Arg334Trp) het
						c.2167+5G>A het
TB785-1 (M)	MA	54	57	Mild (62)	*USH2A*	c.10859; p.(Ile3620Thr) hom
TB898-1 (M)	ASH	70	18	Moderate (76)	*USH2A*	c.12575G>A; p.(Arg4192His) hom
TB1143-1 (F)	EJ	10	30	Mild (34)	*USH2A*	c.784+14389G>T het
						c.7951A>G; p.(Asn2651Asp) het
TB1306-1 (F)	NAJ	20	40	NA	*USH2A*	c.1000C>T; p.(Arg334Trp) hom
TB53-1 (M)	YJ	45	4	Severe–profound (63 y)	*USH1C*	c.1220del; p.(Gly407Glufs*58) hom

Variants refer to the following transcripts: *USH2A*: NM_206933.2 and *USH1C*: NM_005709.3. ASH, Ashkenazi Jewish; EJ, Ethiopian Jewish; F, female; het, heterozygote; HL, hearing loss; hom, homozygote; M, male; MA; Muslim Arab; NA, not available; NAJ, North African Jewish; OJ, Oriental Jewish; RP, retinitis pigmentosa; y, years; YJ, Yemenite Jewish.

## Data Availability

All results obtained are presented in the text, the Figures or the Tables. Raw data can be obtained upon request. The identified pathogenic variants have been submitted to the Leiden Open Variation Database (LOVD) (http://www.lovd.nl, accessed on 4 April 2025) and/or to ClinVar (https://www.ncbi.nlm.nih.gov/clinvar/, accessed on 1 May 2025).

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
