# Peer review of "Phenotypic and Genotypic Characterization of 171 Patients with Syndromic Inherited Retinal Diseases Highlights the Importance of Genetic Testing for Accurate Clinical Diagnosis"

_genes, 2025, doi:10.3390/genes16070745_

Round 1
Reviewer 1 Report
Comments and Suggestions for Authors
The authors present a well-written and comprehensive study detailing the results of several decades of genetic analysis of patients with isolated or syndromic inherited retinal dystrophies. This manuscript provides sufficient background to orient readers and a very thorough, yet concise description of their results. Overall the manuscript will be of value to the broader IRD genetics community. There are only a few extra details that could be added to further improve the manuscript prior to publication.
Introduction
It would be informative for the audience for the authors to include the incidence of the various syndromic forms of IRD listed in their introduction.
Methods
The authors do not list an approving IRB or associated protocol number with their study. They also do not describe if/how informed consent was obtained from their cohort. Could the authors please provide this information in the methods section.
At line 120 and 121 the authors use the abbreviations ES and GS for exome and genome sequencing, however they should use whole exome sequencing (WES) and whole genome sequencing (WGS) to be consistent with the broader field and their own reference (11).
The authors need to list which databases were used to determine allele frequency (gnomAD, and any other if used as the others state they searched public databases for the presence of the TUBB4B variant at line 221). Please also be sure to cite gnomAD using PMID 32461654.
Results
Of the unsolved families (544), how many were syndromic? Have they undergone similarly extensive genetic testing? Do you have any patients that are heterozygous for variants in autosomal recessive genes for syndromic RP? I’m curious as to if you have investigated for the possibility of deep intronic variants or variants within promotor regions which may be able to explain additional cases.
For the patient carrying a homozygous variant in PEX6 (c.1802G>A), did the authors confirm that there was not a deletion of the entire other allele, or a smaller deletion affecting on that region of PEX6? IF the other allele is deleted this could change the interpretation of their findings for phenotype/genotype relationship since the deleted allele would presumably be a full loss of function instead of a hypomorph. Without determining this the authors can’t report that the patient is indeed homozygous for this variant. This is must be completed prior to publication, or the authors must change their statement to likely homozygous. Please either provide data from each parent confirming they are heterozygous for the variant, or present data showing that there is not a deletion of the PEX6 gene in this patient.
Author Response
- It would be informative for the audience for the authors to include the incidence of the various syndromic forms of IRD listed in their introduction.
Answer: Incidence of the various syndromic forms of IRD listed in the introduction is now provided.
- The authors do not list an approving IRB or associated protocol number with their study. They also do not describe if/how informed consent was obtained from their cohort. Could the authors please provide this information in the methods section.
Answer: IRB protocol numbers, as well as a description of how informed consent was obtained, are now provided in the Methods section (sub-section 2.1, 2nd paragraph).
- At line 120 and 121 the authors use the abbreviations ES and GS for exome and genome sequencing, however they should use whole exome sequencing (WES) and whole genome sequencing (WGS) to be consistent with the broader field and their own reference (11).
Answer: The abbreviations ES and GS were changed to WES and WGS (line 149 and throughout the manuscript).
- The authors need to list which databases were used to determine allele frequency (gnomAD, and any other if used as the others state they searched public databases for the presence of the TUBB4B variant at line 221). Please also be sure to cite gnomAD using PMID 32461654.
Answer: Databases used in the study are now listed in the Methods section. PMID 32461654 is cited for gnomAD (sub-section 2.4., first paragraph).
- Of the unsolved families (544), how many were syndromic? Have they undergone similarly extensive genetic testing? Do you have any patients that are heterozygous for variants in autosomal recessive genes for syndromic RP? I’m curious as to if you have investigated for the possibility of deep intronic variants or variants within promotor regions which may be able to explain additional cases.
Answer: The answers to the reviewer’s comments are now provided in the Methods section (sub-section 2.1. lines 147-153) and in the new Supplementary Figure S1.
- For the patient carrying a homozygous variant in PEX6 (c.1802G>A), did the authors confirm that there was not a deletion of the entire other allele, or a smaller deletion affecting on that region of PEX6? IF the other allele is deleted this could change the interpretation of their findings for phenotype/genotype relationship since the deleted allele would presumably be a full loss of function instead of a hypomorph. Without determining this the authors can’t report that the patient is indeed homozygous for this variant. This is must be completed prior to publication, or the authors must change their statement to likely homozygous. Please either provide data from each parent confirming they are heterozygous for the variant, or present data showing that there is not a deletion of the PEX6 gene in this patient.
Answer: Unfortunately, we did not have access to DNA from the patient’s parents, and a quantitative coverage analysis was not performed. Therefore, the patient’s genotype was changed from “homozygous” to “likely homozygous” (line 257). We also mention the possibility of a heterozygous deletion of PEX6 (lines 263-268).
Reviewer 2 Report
Comments and Suggestions for Authors
In their study titled "Phenotypic and genotypic characterization of 171 patients with syndromic inherited retinal diseases highlights the importance of genetic testing for accurate clinical diagnosis," Kulyamzin et al. have presented their cohort of patients with syndromic IRDs and discussed the intricacies of phenotype-genotype correlations with examples of patients with atypical presentations. While their work is a valuable contribution to the field, I have some comments that should be addressed prior to publication:
-More information should be added about other syndromic IRDs to the introduction section that are mentioned later, specifically BBS, JBTS, HMLR, and TUBB4-associated disorders.
-Lines 143-148 should be moved to the methods section under 'subjects'
-In section 3.2, it should be added which type of genetic testing was most commonly used in cases where the initial diagnosis was revised. From the supplementary table, it seems to be exome sequencing, followed by founder testing.
-A paragraph should be added to the discussion section about testing strategies for suspected syndromic IRDs. Is there a "one size fits all" genetic test (ES/GS) or should different approaches be used on a case-by-case basis?
-A paragraph should be added at the end of the discussion section about the strengths and limitations of your study.
-KATNIP is expressed in the kidney according to the Human Protein Atlas. Maybe this information could support the renal involvement of your patient with JBTS26?
-Line 48 'IRD is' should be 'IRDs are'
Author Response
- More information should be added about other syndromic IRDs to the introduction section that are mentioned later, specifically BBS, JBTS, HMLR, and TUBB4-associated disorders.
Answer: More extended information regarding the syndromic IRD forms previously mentioned in the introduction, as well as BBS, JBTS, HMLR2 and TUBB4-associated disorders, is now provided in the Introduction (lines 96-129).
- Lines 143-148 should be moved to the methods section under 'subjects'
Answer: Done (now lines 140-147).
- In section 3.2, it should be added which type of genetic testing was most commonly used in cases where the initial diagnosis was revised.
Answer: The most commonly used type of genetic testing in these cases was WES, as now indicated on line 230.
- A paragraph should be added to the discussion section about testing strategies for suspected syndromic IRDs. Is there a "one size fits all" genetic test (ES/GS) or should different approaches be used on a case-by-case basis?
Answer: A paragraph on this topic was added to the Discussion (lines 414-429).
- A paragraph should be added at the end of the discussion section about the strengths and limitations of your study.
Answer: A paragraph about the strengths and limitations of the study was added at the end of the Discussion (lines 430-444).
- KATNIP is expressed in the kidney according to the Human Protein Atlas. Maybe this information could support the renal involvement of your patient with JBTS26?
Answer: This information is now indicated on lines 406-407.
- Line 48 'IRD is' should be 'IRDs are'
Answer: Done (now on line 53).
Reviewer 3 Report
Comments and Suggestions for Authors
I congratulte the Authors for their perseverance in data collecting. The manuscript is interesting, however, it needs revision.
1. I encourage the Authors to include a few brief points about what was known and what the paper adds.
2. In the Introduction section, it should be emphasized what distinguishes the manuscript from other articles on this topic. The prevalence of the Usher syndrome should be added in the Introduction section.
3. Detailed definition of a family must be included in the Methods section (when were patients considered as the family?
4. Statistical analysis should be added in the Methods section. Otherwise the study has limited scientific value.
5. Demographic data should be presented in the Table.
6. Limitations of the study should be included in the Discussion section.
7. The clinical value of the manuscript should be emphasized in the Conclusion section.
8. References should be expanded
Author Response
- I encourage the Authors to include a few brief points about what was known and what the paper adds.
Answer: We now refer to these issues on the Discussion (lines 430-448).
2. In the Introduction section, it should be emphasized what distinguishes the manuscript from other articles on this topic. The prevalence of the Usher syndrome should be added in the Introduction section.
Answer: This is now emphasized in the Introduction (lines 131-137). The prevalence of Usher syndrome was added to the Introduction (line 77).
- Detailed definition of a family must be included in the Methods section (when were patients considered as the family?)
Answer: A definition of a family is now provided in the Methods section (lines 143-145).
- Statistical analysis should be added in the Methods section. Otherwise the study has limited scientific value.
Answer: I do not understand the reviewer’s comment. The study describes genotypic and phenotypic findings in a cohort of Israeli patients with syndromic IRD. The only type of statistical data included in this manuscript is relative frequency: The relative frequency of different ethnic groups in the cohort (number of families of a each ethnicity out of the total number of families in the cohort); The relative frequency of different phenotypes in the cohort (number of families with each phenotype out of the total number of families in the cohort); The relative frequency of different causative genes in the cohort (number of families with mutations in each causative gene out of the total number of families in the cohort). There are no comparisons between frequencies in different groups, for which you should calculate statistical significance. What type of statistical analysis does the reviewer think should be included here?
- Demographic data should be presented in the Table.
Answer: Demographic data (sex and ethnicity) was added to Table 1.
- Limitations of the study should be included in the Discussion section
Answer: Limitations of the study are now discussed on lines 439-445.
- The clinical value of the manuscript should be emphasized in the Conclusion section.
Answer: The Conclusions section was slightly re-phrased to emphasize the clinical value of the manuscript (lines 450-452).
8. References should be expanded
Answer: References were expanded from 76 to 96.
Round 2
Reviewer 1 Report
Comments and Suggestions for Authors
This reviewer would like to thank the authors for addressing all of my comments and concerns. I have no further comments to add to this revised version. It reads very well and the extra details enhance the overall study. Well done!
Reviewer 3 Report
Comments and Suggestions for Authors
The menus ript has been revised sufficiently